# HBM4EU Occupational Biomonitoring Study on e-Waste—Study Protocol

**DOI:** 10.3390/ijerph182412987

**Published:** 2021-12-09

**Authors:** Paul T. J. Scheepers, Radu Corneliu Duca, Karen S. Galea, Lode Godderis, Emilie Hardy, Lisbeth E. Knudsen, Elizabeth Leese, Henriqueta Louro, Selma Mahiout, Sophie Ndaw, Katrien Poels, Simo P. Porras, Maria J. Silva, Ana Maria Tavares, Jelle Verdonck, Susana Viegas, Tiina Santonen

**Affiliations:** 1Radboud Institute for Health Sciences, Radboudumc, P.O. Box 9101, 6500 HB Nijmegen, The Netherlands; 2Department of Health Protection, Laboratoire National de Santé (LNS), 1 rue Louis Rech, 3555 Luxembourg, Luxembourg; radu.duca@lns.etat.lu (R.C.D.); emilie.hardy@lns.etat.lu (E.H.); 3Centre for Environment and Health, Department of Public Health and Primary Care, KU Leuven (University of Leuven), Kapucijnenvoer 35, 3000 Leuven, Belgium; lode.godderis@kuleuven.be (L.G.); katrien.poels@kuleuven.be (K.P.); jelle.verdonck@kuleuven.be (J.V.); 4Institute of Occupational Medicine (IOM), Edinburgh EH14 4AP, UK; karen.Galea@iom-world.org; 5IDEWE, External Service for Prevention and Protection at Work, Interleuvenlaan 58, 3001 Heverlee, Belgium; 6Department of Public Health, University of Copenhagen, Øster Farimagsgade 5, 1353 Copenhagen, Denmark; liek@sund.ku.dk; 7Health & Safety Executive, Buxton SK17 9JN, UK; liz.leese@hse.gov.uk; 8INSA, National Institute of Health Dr. Ricardo Jorge, Department of Human Genetics, Av. Padre Cruz, 1600-609 Lisbon, Portugal; henriqueta.louro@insa.min-saude.pt (H.L.); m.joao.silva@insa.min-saude.pt (M.J.S.); a.tavares@insa.min-saude.pt (A.M.T.); 9Centre for Toxicogenomics and Human Health, NOVA Medical School, Universidade NOVA de Lisboa, Campo dos Mártires da Pátria 130, 1169-056 Lisbon, Portugal; 10Finnish Institute of Occupational Health, Topeliuksenkatu 41 b, 00250 Helsinki, Finland; selma.mahiout@ttl.fi (S.M.); simo.porras@ttl.fi (S.P.P.); tiina.santonen@ttl.fi (T.S.); 11INRS, French National Research and Safety Institute, 54500 Vandœuvre-lès-Nancy, France; sophie.ndaw@inrs.fr; 12NOVA National School of Public Health, Public Health Research Centre, Universidade NOVA de Lisboa, 1600–560 Lisbon, Portugal; susana.viegas@ensp.unl.pt; 13Comprehensive Health Research Center (CHRC), 1600–560 Lisbon, Portugal

**Keywords:** biomarkers, flame retardants, phthalates, cadmium, chromium, lead, mercury, occupational exposure, recycling

## Abstract

Workers involved in the processing of electronic waste (e-waste) are potentially exposed to toxic chemicals. If exposure occurs, this may result in uptake and potential adverse health effects. Thus, exposure surveillance is an important requirement for health risk management and prevention of occupational disease. Human biomonitoring by measurement of specific biomarkers in body fluids is considered as an effective method of exposure surveillance. The aim of this study is to investigate the internal exposure of workers processing e-waste using a human biomonitoring approach, which will stimulate improved work practices and contribute to raising awareness of potential hazards. This exploratory study in occupational exposures in e-waste processing is part of the European Human Biomonitoring Initiative (HBM4EU). Here we present a study protocol using a cross sectional survey design to study worker’s exposures and compare these to the exposure of subjects preferably employed in the same company but with no known exposure to industrial recycling of e-waste. The present study protocol will be applied in six to eight European countries to ensure standardised data collection. The target population size is 300 exposed and 150 controls. Biomarkers of exposure for the following chemicals will be used: chromium, cadmium and lead in blood and urine; brominated flame retardants and polychlorobiphenyls in blood; mercury, organophosphate flame retardants and phthalates in urine, and chromium, cadmium, lead and mercury in hair. In addition, the following effect biomarkers will be studied: micronuclei, epigenetic, oxidative stress, inflammatory markers and telomere length in blood and metabolomics in urine. Occupational hygiene sampling methods (airborne and settled dust, silicon wristbands and handwipes) and contextual information will be collected to facilitate the interpretation of the biomarker results and discuss exposure mitigating interventions to further reduce exposures if needed. This study protocol can be adapted to future European-wide occupational studies.

## 1. Introduction

Electronic equipment can be found in a wide range of consumer and professional products. This does not only apply to dedicated electronic devices, such as cell phones or computers, but also to electronics integrated in other commodities, such as kitchen appliances and cars. Due to the critical role of these components, their software updates and data security, the electronic hardware needs replacements, leaving behind complete functional units that contain components that could be reused or from which precious metals and polymers could be recycled. Electronic waste or e-waste is defined as [1]: ’various forms of electric and electronic equipment that have ceased to be of value to their users or no longer satisfy their original purpose […] including both “white goods”, such as refrigerators, washing machines, and microwaves and “brown goods”, such as televisions, radios, computers, and cell phones.’ E-waste items are further specified by Baldé and co-workers [2].

Chemicals used in electric and electronic equipment (EEE) consist of metals with impurities from mining and industrially produced technical mixtures of organic chemicals with impurities such as flame retardants and phthalates [3,4,5]. EU regulation on EEE is based on the United Nation’s Agreement on Long-Range Transboundary Air Pollution (CLRTAP) of 1983 [6] and the Stockholm Convention on Persistent Organic Pollutants [7]. In the EU goals of these treatments are achieved with two directives: Directive 2011/65/EU Restriction of Hazardous Substances in Electrical and Electronic Equipment (RoHS) of 2011 [8] aims to prevent the risks to health and the environment through e-waste management specifically targeting metals, flame retardants and plasticizers. This directive contains requirements on effective recyclability of e-waste. With this directive the European Commission also tries to maintain a level playing field for manufacturers and importers of e-waste within the EU. A second directive, Directive 2012/19/EU, Waste from Electrical and Electronic Equipment (WEEE), that was established in 2012 [9] requires EU member states to arrange separate collection, recovery and recycling of e-waste components. Mechanisms are provided to avoid illegal waste exports by making it harder for exporters to disguise illegal shipments of e-waste and harmonise registration and reporting systems across EU member states for enforcement of requirements on e-waste shipments.

Only 35% (3.3 million tons of 9.5 million tons) of used (but in part still functioning) e-waste was processed within the EU. Annually, approximately 400,000 tons of discarded electronics left the EU as part of ‘undocumented mixed exports’ [10]. When taking into account the EU new circular economy policy [11] and the need to enhance the recycling of e-waste [12], the waste management/recycling sector within the EU is expected to grow.

The e-waste stream is complex because it contains many composite materials, such as circuit boards, cathode ray tubes, flat screen monitors, batteries, connectors and transformers, plastic casings and cables [13]. These waste materials contain a broad range of hazardous chemicals, including toxic metals, polybrominated diphenyl ether (PBDE) and organophosphate ester (OPE) flame retardants, phthalates, polychlorobiphenyls (PCBs), hexabromocyclododecanes (HBCDs), polychlorinated dibenzo-p-dioxins (PCDDs), polybrominated dibenzo-p-dioxins (PBDDs) and polychlorinated dibenzofurans (PCDFs) [13,14]. Plastic materials may contain chemicals that were legal at the time they were manufactured, but are now either restricted or banned, such as lead, PCBs, some phthalates and some brominated flame retardants. The recycling of these materials can result in the exposure of workers involved in different steps of the e-waste processing chain, such as collection, sorting, dismantling, shredding and further pre-processing, and purification of waste components for the market of recycled polymer plastics and metals [15].

The European human biomonitoring initiative, HBM4EU (www.hbm4eu.eu/about-hbm4eu) is a European Joint Programme, which aims to harmonise and use human biomonitoring to understand human exposure to chemicals in the environment [16]. Chemical exposures may occur via food consumption and use of consumer products related to potential health risks. HBM4EU includes the use of biomonitoring in the characterisation of the exposure and risks to both the general population and to workers. Occupational exposure to specific chemicals is usually much higher compared to the exposure of the general public, but better understanding exposure in specific occupational settings can support policy actions that can also contribute to preventing exposure of the general public.

Human biomonitoring (HBM) provides important information on the combined exposure via all relevant routes of exposure: inhalation, oral, dermal and hand-to-mouth contact. It usually complements environmental measurements by giving a measure of the internal dose and can inform the efficacy of preventive and protective measures, including personal protective equipment (PPE) [17]. No human biomonitoring guidance has been adopted under the Carcinogens and Mutagens Directive (CMD), although biomonitoring can support the exposure assessment under both REACH and CMD, since it provides an estimate of the internal dose and, for example, on the effectiveness of respiratory protection to reduce exposure.

HBM4EU can support the development of sustainable practices in e-waste management by providing suitable methods for exposure assessment that demonstrate the need for development of sound practices in professional processing of e-waste. This may prevent e-waste from being dumped in, and also outside, European countries and would support more sustainable processing of this waste stream in line with the Basel Convention on the Control of Transboundary Movement of Hazardous Wastes and their Disposal [18]. A recent literature review by Arain and Neitzel (2019) [19] showed that, so far, occupational exposures occurring in this sector were only studied in one European country [20]. We identified a second study also reporting on the use of biomonitoring to assess the level of exposure to metals [21]. These studies together reported higher chromium, cobalt, indium, lead, and mercury concentrations in the urine and/or plasma of the recycling workers, compared to controls.

The current protocol describes an EU-wide exploratory effort to characterize the working conditions in the e-waste processing sector and the associated exposure to chemicals. For this study our focus will be on the occupational health and safety aspects. The primary objective of this project is to study the internal exposure of workers processing e-waste using a human biomonitoring approach, contributing to raising awareness of potential hazards. In a partnership with the recycling sector in Europe, HBM4EU can help to ensure the sustainable processing of e-waste. This may result in an increase in the share of processing e-waste that Europe is producing, using Europe’s own processing capability, instead of exporting e-waste [13]. By conducting an HBM study, we hope to contribute to stimulating good work practices that will lead to further improved protection of the worker’s health from the risk of exposure to toxic chemicals, including that of combined exposures.

For the HBM4EU e-waste study, the following secondary objective(s) have been formulated:Identify the most relevant chemicals in e-waste processing and use available knowledge developed and available in HBM4EU to support an exposure study.Develop a study protocol, standard operating procedures (SOP), information materials and informed consent forms and documentation for ethics approval in each of the participating member states.Set up a collaboration with those labs that could support the analysis of the most relevant biomarkers in body tissues that can be obtained.Collaborate with employers and employees of companies in the public and private sectors to collect the biological specimens for the HBM surveys.Implement the HBM study with sufficient supportive external exposure measurements and contextual data allowing the identification of opportunities for further improvements of occupational health practice and herewith address questions and concerns that employers and employees might have.

## 2. Study Design

The present protocol describes a cross-sectional study design that is expected to be implemented in six to eight European countries. To achieve the aim of the study, it is important that a wide range of industrial recycling activities are covered and that the study population represents a wide range of work practices. The results will then reflect real life exposures in the participating European countries. 

E-waste processing is often described in a stepwise process where different parties work together in a chain: general waste is collected and treated by a network of service providers in the market of general (household) waste. Other parties may also buy/accept/treat specific (industrial) waste streams for processing. They provide services for collection, transport, sorting and processing. E-waste is a part of this waste stream for which additional services have been developed for specialised treatment. A very generic description is given based on activities that can be identified as part of the e-waste processing chain in Table 1 [22,23,24].

Some of these activities may be provided in one plant at one site (e.g., category 1–3) but companies accepting large waste streams for incineration often do not have the facilities to process the e-waste on the same location and collaborate with other companies for further processing. Usually, they offer these waste streams to companies who specialise in further processing to refurbish recycled products, or re-use as feedstock for other industrial sectors. The further processing may be done by small companies. This can also involve large multinationals with activities in polymers or metal alloys for whom e-waste may be a side activity.

### 2.1. Chemicals of Interest

This occupational biomonitoring study on e-waste will focus on a range of chemicals identified on the first and second list of HBM4EU priority substances, for which biomonitoring methods have been developed and tested in multiple laboratories as part of HBM4EU and may also include chemicals for which this process is still ongoing. To determine the most relevant chemicals to include in our survey we used previous studies conducted, applying both occupational hygiene measurements [25,26,27,28] and biological monitoring approaches [20,21]. We also used reviews that describe the health impact of e-waste processing [14,29,30,31]. Additionally, we consulted some published technical reports from the e-waste processing industry for a good understanding of the technical aspects of e-waste processing [22,23,24]. Based on this exploration the following priorities emerged regarding chemicals of interest (see Table 2).

#### 2.1.1. Metals

In the processing of e-waste, the recovery of precious metals provides opportunities for reuse [13]. Some of these metals raise concern for potential harmful effects to human health and the environment. Workers of waste processing facilities are at risk for inhalation and skin exposure with potential health implications. In the current protocol, selected metals will be analysed in inhalable and respirable dust fractions and in settled dust. For this we will analyse chromium, cadmium, lead and mercury in urine and hair and chromium, cadmium and lead in blood (see Table 2). The analysis of cadmium in blood and urine and chromium in blood, serum and urine were supported in [32]. To explore primary and secondary sources and routes of exposure and for better interpretation of the biomonitoring outcome of inhalable and respirable dust, settled dust and hand wipes will be analysed for the above-mentioned metals.

#### 2.1.2. Brominated Flame Retardants (BFR) and Organophosphate Flame Retardants (OPFR)

Circuit boards are sprayed with flame retardants when dismantling and when shredding casings for electronic equipment it is likely that dust with high flame retardant content is released in the air and that settled dust will also contain considerable flame retardant levels [28,33] that may lead to enhanced inhalation and skin exposure in recycling facilities [27]. Exposure assessment is well-established and based on the analysis of wrist bands and biomonitoring by analysis of blood serum for BFRs [34] and urine for OPFRs [35] (Table 2). From the BFRs and OPFRs we will select the most relevant chemicals based on a pre-screening effort that can provide informed decisions for additional analyses. Analysis of flame retardants in blood serum (BDE-47, BDE-153, BDE-209, α-HBCD, γ-HBCD, anti-DP, syn-DP, TBBPA, DBDPE, 2,4,6-TBP) and four OPFRs metabolites in urine were supported in the HBM4EU QA/QC programme [36].

#### 2.1.3. Phthalates

Electronic equipment is often part of consumer and professional products that contain polymers and plastic coatings that contain plasticizers [37,38,39]. Dismantling and shredding of e-waste will therefore lead to the production of dusts that contain phthalates that may lead to inhalation and dermal exposure. Methods for biomonitoring of exposure to phthalates will be based on urinary analysis of biomarkers (MEP, MBzP, MiBP, MnBP, MCHP, MnPeP, MEHP, 5OH-MEHP, 5oxo-MEHP, 5cx-MEPP, MnOP, OH-MiNP, cx-MiNP, OH-MiDP, cx-MiDP) targeted in the QA/QC programme of HBM4EU and can be applied to workers’ exposures [32]. The link to the working environment will be made by analysis of phthalates in settled dust and wristbands (Table 2).

### 2.2. Human Biological Monitoring

We will use exposure biomarkers and biomarkers of effect as defined by Zielhuis and Henderson [40]. The exposure biomarkers reflect the uptake of specific chemicals of interest via different sources and routes of uptake integrated over time. Biomarkers of effect describe a wide range of early biological responses that may be associated with the overall chemical exposure encountered but are usually less specific to a chemical species.

Most of the biomarkers of exposure and effect will be determined from both exposed workers and controls in one workweek (Table 2). Controls typically work in the same companies e.g., working in offices and performing administrative tasks on the same industrial site as the exposure workers, but have no known exposure to emissions from e-waste processing (within the e-waste industry). Exposure in office workers can still be elevated from a background in the general population due to undetected sources and routes of exposure dependent on e.g., open air connections between production and office workspace and some facilities used by both groups of workers [41,42]. Regarding the relevance of background exposures for metals, flame retardants and phthalates we will (to a limited degree) also include office workers from employers in industries other than e-waste processing (outside the e-waste industry) [42]. Environmental monitoring (as specified in Table 2) will be performed in exposed workers only.

Chromium and mercury will be measured in different biological media to inform exposure assessment in different time windows. With a half-life of excretion of 2–3 days for chromium in urine and 1.5 days in plasma, the results will primarily reflect recent exposure to both chromium III and chromium VI [43,44], whereas the half-life of chromium in red blood cells of 60 days reflects long-term exposure to specifically chromium VI [45]. With a half-life of 1.2 and 10.5 days, mercury in blood will primarily reflect recent exposure, whereas mercury in urine may reflect exposure over weeks to months [46].

Mercury in urine reflects mostly the exposure to inorganic mercury whereas blood total mercury reflects exposure to both inorganic and organic mercury species unless speciation is made. This is the reason why we focus on U-Hg measurements in this study. In hair the metal content will also likely reflect uptake over a long period of time [46,47]. The half-life of lead from blood was reported to be 28–36 days [48,49] and the urinary excretion half-life in humans is not reported.

Urine is collected on the first and last day of the workweek. The first sample represents the baseline (after 48 h off work) and is collected before the start of the first shift of the workweek (pre-shift) on the first day of the workweek. This morning urine sample is collected at home as the first void after awakening, with participants asked to take this urine sample to their workplace. The second (post-shift, end-of-the week) reflects the contribution of the exposure during the workweek and is collected immediately after the end of the shift before going home. Workers are instructed to remove overalls and wash their hands before collection so to avoid contamination of the urine sample. Collection of blood, buccal swipe and hair is combined with taking the questionnaire by interview (see Section 2.4 for more information on this) on day 3–5 of the workweek (as the timing is less critical). For the participants in the control group, only blood and urine samples will be collected to assess a baseline level in a comparable group of workers with no known exposure as a reference. 

Additionally, the use of effect biomarkers is of utmost importance to establish a relationship between the exposure to chemicals present in e-waste and its human health impact, given that they comprise sensitive endpoints reflecting early biochemical/subclinical changes before the onset of disease. For the hazardous properties of the chemical exposures a series of effect biomarkers was selected (Table 3). 

### 2.3. Occupational Hygiene Measurements

In order to get an overview of the concentration of potential e-waste chemicals at the workplace and to help identify the principal exposure routes, personal air samples will be collected of settled dust, airborne dust and dermal exposures. Deposits of settled dust will be collected by use of a vacuum cleaner in a designated area in a production environment.

Inhalation exposure is important when considering the risk of respiratory disease. Dermal exposure and oral uptake affect the whole-body burden and may increase the risk for effects on the gastrointestinal system. Inhalable and respirable air samples are collected during at least 6 h according to EN689 [50]. Every workday at least one air sample will be collected within each similar exposure group (SEG). 

Wipe samples from the dominant hand of the workers will also be collected, since skin contamination may represent a significant exposure route to chemicals, due to dermal absorption or gastrointestinal absorption as a result of hand-to-mouth contact. Three to five hand wipe samples (depending on number of breaks) will be collected on the same workday (the choice of the day is not critical).

### 2.4. Contextual Data Collection

Generic questionnaires were developed for occupational studies and have been used as a basis and adapted to the purposes of the current study (Table 4). The first questionnaire (Q1) focuses on characteristics of the workplace with the aim to collect general information on the operating conditions and risk management measures (RMMs). This questionnaire can be completed by the researcher (occupational hygienist) after consultation of the site manager of the company. In a second short questionnaire (Q2) the workers will be asked to confirm the samples collected during the shift and the related task and work location. A third extended questionnaire (Q3) was developed for individual study participants and includes detailed questions on the job description, the specific work tasks, relevant co-exposure from potential other sources outside work and RMMs, including the availability and use of PPE. Q3 will be interview-led by the researcher to make sure that the workers understand and answer to all the questions. During the shift, the researchers will visit the participants at their workplaces at regular intervals to collect contextual data on the work practices with special attention also being given to ensuring the correct function of the air sampling equipment, e.g., correct positioning of the sampling heads in the breathing zone and verification of the flow calibration of the pumps. This will also provide an opportunity to respond to questions that workers may have.

### 2.5. Implementation of the Data Collection

All samples and data will be collected during an uninterrupted sequence of 4–5 shifts (Table 5). When the researchers visit the workplace, they will collect data that will later be used to interpret the measurements and for modelling of the exposure conditions. More specifically they will collect data on:the specific tasks that are carried out with special attention to tasks that may cause high/peak exposures;the type and use of PPE such as gloves, goggles and respiratory protection;eating, drinking and smoking and other personal hygiene aspects at the workplace;occurrence of direct unprotected skin contact with chemicals/waste and/or potentially contaminated surfaces;occurrence of unexpected events such as spills which could lead to exposure.

These observations will be registered in a check list for the standard items and a sample log to report on the performance characteristics of the sampling equipment (such as flow calibration) and on the occurrence of events that could have a direct or indirect influence on the measurements. All events will be registered with a worker participant code, workplace location and clock time. For comparison of the occupational hygiene measurements with exposure, estimates will be generated with the Advanced REACH Tool via the TREXMO model [51].

## 3. Study Population

### 3.1. Definitions of Index and Control Groups

The target population consists of workers employed by companies involved in the processing of e-waste including: sorting, dismantling, shredding and pre-processing and metal as well as non-metal processing (see Table 1). The exact techniques used in the recycling industries will be specified when collecting contextual information. The index population consists of production workers actively involved in e-waste processing and in direct contact with the materials and process emissions. A control group will be studied consisting of individuals neither involved in e-waste processing nor in other activities with known occupational exposure to the chemicals of interest. Only if companies have insufficient workers in the latter group, an attempt will be made to find suitable controls in other occupations with no known exposures to the chemicals of interest (non-industrial controls). The non-industrial controls involved will be from the same geographical region as exposed workers. Samples are expected to be collected from the following countries: Belgium, Finland, Latvia, Luxembourg, The Netherlands, Poland, Portugal and UK.

Eligible participants need to be aged eighteen years or older on the day of recruitment, and need to have an employment contract by the company or a contractor.

### 3.2. Sample Size Calculation

The purpose of the sample size calculation is to justify that the study sample will be of sufficient size to expect statistically meaningful differences based on intra and inter individual differences in biomarker levels in previous studies of limited size. Our study protocol describes a first exploratory study in a limited number of countries in a small number of industries per country. We do not expect that we can capture the variability of all work practices in each of the participating countries to reflect all e-waste management practices. Our study sample will lead to a representative subset of work practices. We can only describe the level of occupational health and safety standards that we find in our company selection. We do not see this as a problem for our study protocol because our aim is to discuss the need for further improvement of work conditions in the e-waste management sector. In that way we expect that all companies (also the companies with lower occupational health and safety standards) can benefit from the results of our study.

The studies by Julander and co-workers and Gerding and co-workers are currently the only identified published studies in Europe reporting on the use of human biomonitoring in e-waste workers [20,21]. Julander and co-workers only reported on median and range which makes it difficult to use these data in a power-calculation [20]. Recently, Gerding and co-workers performed a biomonitoring study in e-waste processing in Germany, but the authors did not report any differences in metal urinary excretion in 51 production workers and 20 controls except for mercury [21]. For chromium, Scheepers and co-workers reported on a human biomonitoring study performed in the Netherlands, using a similar, cross-sectional design [52]. A population of 28 stainless steel (SS) and 19 mild steel (MS) welders was studied and compared to 20 controls for different biomarkers, including urine and plasma. Edmé and co-workers did a similar study in France in 1993 [53] They compared 115 SS welders and 32 MS welders with 26 control subjects for all biomarkers. Based on these studies some power calculations were performed for the different biomarkers (Table 6).

For the HBM4EU e-waste study, the small differences of MS welders vs. controls would require study groups sizes of 311 at 80% power and 417 at 90%. It can reasonably be assumed that the worker-to-worker variability will be similar in the HBM4EU e-waste study. Post-shift urinary excretion data from di(2-ethylhexyl) phthalate (DEHP) and phthalic acid were taken from an exposure study among workers of a chemical plant producing DEHP from phthalic acid and 2-ethylhexanol [55]. For these data and other endpoints like mercury in hair [54] the required number of exposed workers and controls are similar to those indicated for welders. Overall, the target population size for this project, assuming three to four companies from two to four categories of e-waste processing, is presented in Table 7.

On a national level, a sample size of 50 for exposed and of 25 for controls, and within a category of e-waste processing a sample size of 60 for exposed and of 30 for controls across all participating countries would be sufficient to find statistically significant results when comparing biomarker levels between groups of exposed workers (similar to the higher exposed groups, e.g., SS welders) and controls with reasonable power of 80%. On a European level, this study is expected to be performed in 6–8 countries, which would lead to at least 300 workers and 150 control subjects. Based on this sample size it is likely that differences between workers groups (even with low exposure) compared to biomarker levels in controls would be detectable with reasonable power.

## 4. Ethics and Data Protection

The study will be conducted in line with regulations of the European Commission and the current General Data Protection Regulation (GDPR). The study also adheres to the principles of the Declaration of Helsinki (64th WMA General Assembly, Fortaleza, Brazil, October 2013) and is in accordance with national legislation requirements of each participating country. The HBM4EU Legal and Ethics Policy [56] will be followed. We will also adhere to the HBM4EU Material and associated Data Transfer Agreement (MDTA) [57]. Because of the occupational setting where study participants of both exposed and control groups are in a dependent position of their employer, extra attention is given to allow a free decision about participation, i.e., a decision without any direct influence of the employer. We also build on prior experience from an earlier HBM4EU study on chromate [58].

With the employer only non-personally identifiable confidential information (on a group level) will be shared. In smaller groups of workers and small companies, researchers should be aware that personally identifiable information (PII) should not be released as this could lead to a breach of privacy of the individual participant and violate confidentiality. We will conduct the study with respect for all groups in society regardless of race, ethnicity, religion, and culture, and with respect for and awareness of gender or other significant social differences as to avoid a situation of unnecessary stigmatization [56].

Any information presented on a European level will be in a strictly non-identifiable format so that no one will be able to identify anyone (or any company) who took part in the study. The results of the study will be reported within the reporting policy of the HBM4EU project, and a general report will be publicly accessible via the HBM4EU web site [59]. The results will be published as scientific publication(s) in peer-reviewed scientific journals, and presented at scientific meetings. No personally identifiable results will be presented.

## 5. Recruitment and Consent

The recruitment of workers will be conducted in companies who agreed to participate in the study as per employer consent (as described in Appendix A). Once companies have agreed to participate, an invitation letter will be provided to workers who belong to the target group (workers with known exposure to e-waste components and workers with no known occupational exposure to e-waste components).

Workers eligible for participation as production worker or control worker (referred to as ‘worker’ in the text below) will be identified by their task description. In large companies, potential participants will also be identified using the principle of ‘similar exposure groups’ (SEG), described in European standard EN689 [50]. 

The procedure for recruitment will be as follows:A call for volunteers is issued to workers. Those expressing interest receive the participant information about the study and an informed consent form. English versions of both forms are provided in Appendix A.The researcher meets the workers in person at the company to discuss any questions related to the research project and their participation in this project.Those wishing to participate then give written consent prior to the start of the study and the researcher co-signs this form.Subjects can leave the study at any time for any reason if they wish to do so without any consequences. In the informed consent we will ask study participants if we may still use biological samples that have been collected before the decision to discontinue participation. If the participants would not like to know their results, they can indicate this.Specific criteria for withdrawal (if applicable) are
The participant wants to discontinue participationEmployment endsWorker has a leave of absence due to illness or personal circumstancesIf a participant has withdrawn from the study, another worker may be recruited to a maximum of 40% of the target for both controls and exposed.

## 6. Methods for Sample Collection and Analysis

All used methods for sample collection and analysis have been described in standard operation procedures (SOPs) (Table 8).

As an example, Figure 1 depicts the sample distribution after peripheral blood collection in order to proceed with the several analyses to be performed. Below, the primary and secondary study parameters are listed as well as other study parameters and occupational hygiene measurements.

### 6.1. Primary Study Parameters/Endpoints

The exposure biomarkers are (Table 2):chromium, cadmium and lead in bloodbrominated flame retardants and polychlorobiphenyls in bloodchromium, cadmium, lead and mercury in urineorganophosphate flame retardants in urinephthalates in urinemetals in hair.

### 6.2. Secondary Study Parameters/Endpoints

In addition to the aforementioned exposure biomarkers, the following effect biomarkers will be studied (Table 3):micronuclei frequenciesepigenetic markersoxidative stress markersinflammatory markerstelomere lengthmetabolomics.

### 6.3. Other Study Parameters

Creatinine in urine, a metabolite of creatine for urine density adjustment

### 6.4. Occupational Hygiene Measurements

Inhalable and respirable dust in breathing zone by personal air samplingSettled dust for metals flame retardants and phthalatesHand wipes for metals analysisWrist band for analysis of phthalates and flame retardants.

## 7. Data Analysis and Reporting

All collected data will be pseudonymised before any treatment by replacing participant and company names with a code and protecting all electronic and paper records from unauthorised access. Only the coordinating researcher of the study and the occupational physician will have access to these data. For all communications related to the study outcomes a communication plan was developed and described in a SOP specifically dedicated to this purpose (Table 8). 

With the employer only non-identifiable information (on a group level) will be shared. In smaller groups of workers and small companies, researchers should be aware that personally identifiable information (PII) should not be used as this could lead to a breach of privacy of the individual participant [56]. The participating companies will receive a report considering environmental and biomonitoring results on an aggregated level (no individual results will be provided to the employer). The study participants will receive a summary of the aggregated results on a company level. In some countries the individual study results will be added to the medical file, whereas in other countries the study participants receive their own individual results, including some basic interpretation of what the results indicate, unless they have specifically indicated on their informed consent form that they do not want to know their results. If study participants do not receive their individual results automatically, they can request this access from the researcher or occupational physician. Only if they have agreed to it on the informed consent form will study outcomes of general medical interest be shared with the general practitioner. For reporting of study outcomes to participants, the study leaders will adhere to the right to know and the right not to know as an overarching principle.

Any information presented on a European level will be in a strictly non-identifiable format so that no one will be able to identify anyone (or any company) who took part in the study. The results of the study will be reported within the reporting policy of the HBM4EU project, and a general report will be publicly accessible via the HBM4EU web site [59]. The results will be published as scientific publication(s) in peer-reviewed scientific journals, and presented at scientific meetings.

The samples and data will be stored for 5 years after completion of the project; blood and urine samples will be stored in −18 °C or −80 °C freezers. This study is part of the HBM4EU project that was started in January 2017 and will be completed in 2022. Completion of the project is defined as the moment when manuscripts describing the project results are accepted for publication. 

## 8. Conclusions

We present here the protocol of a cross-sectional study of worker’s exposures in industrial recycling of e-waste. The present study will be carried out in six to eight European countries according to harmonised and standardised procedures for recruitment of participants, collection of the industrial hygiene and biological samples and of contextual data to support the interpretation of our results. The target population size of 300 exposed and 150 controls should be sufficient to answer our research questions. This is a first exploratory study in this industrial sector in Europe. We expect that the study will increase the knowledge and understanding of how work practices translate to exposures.

## Figures and Tables

**Figure 1 ijerph-18-12987-f001:**
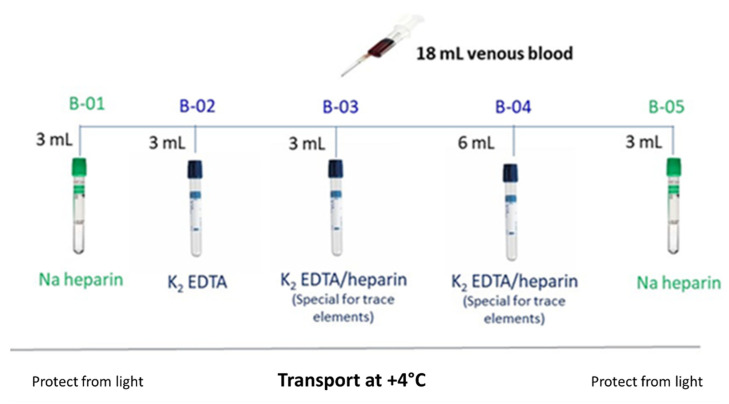
Venous blood sample collection.

**Table 1 ijerph-18-12987-t001:** Categories of e-waste processing activities.

Category	Title	Description
1	Sorting	Sorting e-waste from household and industrial waste streams (by hand or semi-automated)
2	Dismantling	Split casings from electronic components such as circuit boards and batteries (often by hand)
3	Shredding and pre-processing	e.g., on a belt by electrostatic, density, magnetism, colour separation, etc.
4a	Metal processing	Melting metals for re-use as feedstock
4b	Polymer processing	Polymer processing to a granulated material for re-use as feedstock

**Table 2 ijerph-18-12987-t002:** Environmental and biological samples for determination of exposure biomarkers.

Type of Sample	Sample Type	Chemicals of Interest	Time of Sampling for Workers	Number of Samples
Environmental	Inhalable and respirable dust	Chromium, cadmium, lead and mercury	During at least 6 h of the work shift ^a^	>3
	Settled dust	Chromium, cadmium, lead and mercury	During at least one shift	4–5
	Hand wipe	Chromium, cadmium, lead and mercury	Several including pre-shift/post shift and during breaks during one shift ^a^	3–5
	Wrist band	Flame retardants and phthalates	Same wrist band will be worn during work hours for five workdays ^a^	1
Biological	Urine	Chromium, cadmium, lead, mercury, organophosphate flame retardants and phthalates	Pre-shift at the beginning and post-shift, end of the workweek	2
	Blood	Chromium, cadmium, lead, brominated flame retardants and polychlorobiphenyls	Sample collection in the second half of the workweek	1
	Hair	Chromium, cadmium, lead, mercury	Morning preferably before start of the workweek	1

**Table 3 ijerph-18-12987-t003:** Biological samples for determination of effect biomarkers.

Sample Type	Sample Fraction	Effect Parameter
Blood	Peripheral blood lymphocytes	Epigenetic markers in DNA
		Telomere length in DNA
	Peripheral blood lymphocytes and reticulocytes	Micronuclei (frequencies)
	Plasma	Oxidative stress markers in blood
		Inflammatory markers in blood
Buccal swap	Differentiated mononucleated cells from the buccal epithelium	Differentiated mononucleated cells are scored for nuclear alterations (micronuclei and buds). Evaluation of frequencies of basal cells, differentiated cells, and cells with anomalies associated with cell death
Urine	Not applicable	Metabolomics in urine

**Table 4 ijerph-18-12987-t004:** Questionnaires and questionnaire items for determination of contextual information.

Questionnaire	Interview with	When	Items
Q1	Site manager	Before sample collection starts	Company information
			Type and amount of e-waste processed
			Training of workers on hazardous materials
			Operational conditions and risk management measures
			Previous measurements
			Hygiene facilities and procedures
Q2	Worker	Every shift	Sample types collected during the shift with dates, times and work locations
Q3	Worker	Single time	Personal characteristics
			Task-related information
			Domestic situation
			Smoking habits and smoking history (covering both tobacco use and e-cigarettes)
			Drinking of alcoholic beverages
			Dietary habits and use of food supplements
			Orthopaedic or orthodontic implants and dental fillings
			Recreational activities
			COVID-19—history and vaccination status
			Occupational history
			Job description of current tasks in e-waste processing
			Instructions and use of PPE
			Type of ventilation

**Table 5 ijerph-18-12987-t005:** Times and days of sample and data collection for a participant from the exposed group. If sample collection is scheduled on more days, it means that the exact day is not critical. Sample types and numbers of samples per worker are given below in parenthesis: A1–A3 = air sample (inhalable/respirable dust) (*n* ≥ 3 within each SEG); B = blood sample (*n* = 1); BS = buccal swipe (*n* = 1); H1–H5 = hand wipes (*n* = 3–5 preferably on the day of post-shift urine collection); HR = hair sample (*n* = 1); Pre = before the shift on the first day of the workweek; Post = after the shift on the last day of the workweek; U1 = pre-shift urine sample (*n* = 1); U2 = post-shift urine sample (*n* = 1); Q1–Q4 = questionnaires by interview; W = the same wrist band is worn every workday during work hours (*n* = 1).

Workday	Pre	Shift-1	Shift-2	Shift 3–5	Post
Questionnaires	Q1	Q2	Q2	Q2, Q3	
Air sample		A1	A2	A3	
Hand wipe				H1–H5	
Urine ^a^	U1				U2
Blood ^a^				B	
Hair ^a^				HR	
Buccal swipe ^a^				BS	
Wrist band		W	W	W	

^a^ Controls.

**Table 6 ijerph-18-12987-t006:** Power calculation with an alpha of 0.05 for comparison of workers with controls for chemical exposures relevant to e-waste. The enrolment ratio is 2:1 (exposed:controls).

Biomarker	Groups Compared	Worker Mean (sd)	Control Mean (sd)	Study Group Size (Workers)		Reference
				80% Power	90% Power	
Chromium in urine (µg/g crea)	MS welders vs. controls	0.38 (1.2)	0.05 (0.31)	311	417	[52]
	SS welders vs. controls	1.20 (1.2)	0.05 (0.31)	24	33	
	MS welders vs. controls	2.4 (2.3)	0.5 (2.1)	35	46	[53]
	SS welders vs. controls	6.2 (3.4)	0.5 (2.1)	8	11	
Chromium in whole blood (µg/L)	MS welders vs. controls	1.8 (2.3)	1.2 (1.7)	346	463	
	SS welders vs. controls	3.6 (2.3)	1.2 (1.7)	22	29	
Mercury in hair (µg/g creat)	Workers vs. controls	0.3 (0.2)	0.1 (0.1)	36	48	[54]
Di(2-ethylhexyl)-phthalate in urine (pmol/g creat)	Highly exposed workers vs. low exposed workers	9.9 (8.7)	5.9 (4.8)	51	69	[55]

MS = mild steel; SS = stainless steel.

**Table 7 ijerph-18-12987-t007:** Target number of workers and controls to be recruited in total, by category and by participating company.

Level	Study Group	Total	By Category of Processing ^a^		By Participating Company/Site ^b^	
		75	Min 2	Max 5	Min 3	Max 4
Per country	Workers	50	25	10	18–19	12–13
	Controls ^d^	25	12–13	5	9–10	6–7
Total ^c^	Workers	300	150	60	100	75
	Controls ^d^	150	75	30	50	37–38

^a^ See Table 1 for categories of e-waste processing; ^b^ Ranges of 2–5 and 3–4 are used here to indicate that both small and medium enterprises and large industries will be included in both small- and large-scale processing. Some flexibility is required here because of situations that may vary from country to country and by type of processing; ^c^ The number of participating countries is expected to be higher than the number of 6 that was used for this calculation; ^d^ Office workers within and outside e-waste processing (see Section 3.1).

**Table 8 ijerph-18-12987-t008:** Standard operation procedures (SOPs) prepared for this study ^a^.

No.	Title	Appendix A
1	Selection of participants and recruitment, information to the participants, informed consent	Appendix A
2	Completion of company and worker questionnaires	Appendix A
3	Blood sampling, including sample storage and transfer	Appendix A
4	Urine sampling, including sample storage and transfer	Appendix A
5	Settled dust	Appendix A
6	Air sampling of inhalable and respirable dust fraction	Appendix A
7	Obtaining hair samples	Appendix A
8	Dermal sampling using wipes and wristbands	Appendix A
9	Buccal cells sampling including sample storage and transfer	Appendix A
10	Comparing occupational hygiene measurements with exposure estimates generated using the Advanced REACH Tool via the TREXMO model	Appendix A
11	Communication plan for the occupational studies	Appendix A

^a^ Also online available at: https://www.hbm4eu.eu/mdocs-posts/hbm4eu-occupational-e-waste-study-standard-operating-procedures-sops/ (accessed on 28 November 2021).

## Data Availability

No new data were created or analyzed in this study. Data sharing is not applicable to this article.

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
