# Peer review of "HBM4EU Occupational Biomonitoring Study on e-Waste—Study Protocol"

_ijerph, 2021, doi:10.3390/ijerph182412987_

Round 1

Reviewer 1 Report

This is an interesting approach and I am sure a lot of researchers in the e-waste recycling sector are highly curious about the results that will be generated, in particular since data of the formal e-waste sector are missing i Europe. However, there are some major issues that need to be addressed:

  1. Why do the authors do not assess nickel and (inorganic) arsenic? I think these metals are of particular interest since an additional occupational exposure is to be expected in e-waste workers.
  2. An overview about the half-lives of the substances that are monitored and an explanation why a certain substance is measured in different biological materials, e.g. mercury in blood, urine and hair might be helpful.
  3. A weekly hair analysis, e.g. for cadmium is not really clear for me, in particular regarding the half-live. Why is that necessary?
  4. The sample size calculation needs to be explained in more detail. From my point of view, there will be country and company specific influences. I can only speak for Germany, but we have a huge varity in terms of occupational health and safety measures depending on the company. I do not think that HBM in only 50 persones will give a representative country and company-wide overview.
  5. How do the authors consider a selection bias? I am quite sure that they will have less problems in recuriting companies with high occupational health and safety standards.
  6. A small paragraph concerning quality assurance might be helpful. Are there round robin analyses between the different laboratories to confirm the HBM quality?

Author Response

This is an interesting approach and I am sure a lot of researchers in the e-waste recycling sector are highly curious about the results that will be generated, in particular since data of the formal ewaste sector are missing i Europe. However, there are some major issues that need to be addressed:

  1. Why do the authors do not assess nickel and (inorganic) arsenic? I think these metals are of particular interest since an additional occupational exposure is to be expected in e-waste workers.

Author’s response:

The reviewer is correct that there will be additional metals (and other hazardous substances) of concern to e-waste workers.  We also hope that the reviewer appreciates that it is not possible for the project team to address exposures to all of these within the one study.

As explained in Section 2.1 of the manuscript the selection of the substances to be included in the programme of work was based on the following (lines 188-195 in the revision):

  1. First and second list of HBM4EU priority substances, for which biomonitoring methods have been developed and tested in multiple laboratories as part of HBM4EU and may also include chemicals for which this process is still ongoing.
  2. Review of previous studies applying both occupational hygiene measurements and biological monitoring approaches to identify the most relevant chemicals.
  3. Used reviews that describe the health impact of e-waste processing.

Finally, we consulted some published technical reports from the e-waste processing industry for a good understanding of the technical aspects of the e-waste processing. We therefore consider that our selection is sufficiently justified and no changes have been made to the manuscript in response to this comment.

  1. An overview about the half-lives of the substances that are monitored and an explanation why a certain substance is measured in different biological materials, e.g. mercury in blood, urine and hair might be helpful.

Author’s response:

The biological matrix and type of biomarker provides additional information such as demonstrated in the HBM4EU Chromates study (reference 41 in the revision). This is also the case for mercury. We have added the following clarification in lines 265-282 of the revision: “Chromium, mercury and lead will be measured in different biological media to inform exposure assessment in different time windows. With a half-life of excretion of 2-3 days chromium in urine and 1.5 days in plasma will primarily reflect recent exposure to both chromium III and chromium VI [42-43], whereas the half-life of chromium in red blood cells of 4 months reflects long-term exposure to specifically chromium VI [44]. With a half-life of 1.2 and 10.5 days, total mercury in blood reflects recent exposure to both inorganic and organic mercury species whereas mercury in urine reflects mostly the exposure to inorganic mercury over weeks to months [45]. This explains why we prefer urinary mercury measurements in this study. In hair the metal content will also likely reflect uptake over a long period of time [46]. The half-life of lead from blood was reported to be 28-36 days [47-48] and the urinary excretion half-life in humans is not reported.”

  1. A weekly hair analysis, e.g. for cadmium is not really clear for me, in particular regarding the half-live. Why is that necessary?

Author’s response:

We use a cross-sectional study design and plan to recruit workers for one workweek’s observation and measurement. This is clarified in lines 250-251 in the revision of the manuscript.

  1. The sample size calculation needs to be explained in more detail. From my point of view, there will be country and company specific influences. I can only speak for Germany, but we have a huge variety in terms of occupational health and safety measures depending on the company. I do not think that HBM in only 50 persons will give a representative country and company-wide overview.

Author’s response:

The reviewer’s is right to indicate that we should further clarify the sample size calculation. We propose to add the following explanation to section 3.2 on sample size calculation (lines 409-415): “The purpose of the sample size calculation is to justify that the study sample will be of sufficient size to expect statistically meaningful differences based on intra and inter-individual differences in biomarkers levels in previous studies of limited size. Our study protocol describes a first exploratory study in a limited number of countries in a small number of industries per country. We do not expect that we can capture the variability of all work practices in each of the participating countries to reflect all e-waste management practices.”

  1. How do the authors consider a selection bias? I am quite sure that they will have less problems in recruiting companies with high occupational health and safety standards.

Author’s response:

The reviewer is right that the recruitment companies to participate will certainly lead to some selection bias as randomization is not feasible. We have added the following clarification in section 3.2 (lines 415-421) of the revision: Our study sample will lead to a representative subset of work practices. We can only describe the level of occupational health and safety standards that we find in our company selection. We do not see this as a problem for our study protocol because our aim is to discuss the need for further improvement of work conditions in the e-waste management sector. In that way we expect that all companies (also the companies with less high occupational health and safety standards) can benefit from the results of our study.”

  1. A small paragraph concerning quality assurance might be helpful. Are there round robin analyses between the different laboratories to confirm the HBM quality?

Author’s response:

Thank you for this suggestion. We already referred to the inter-laboratory QC/QA scheme that is part of the HBM4EU effort and described in some previous published studies that we refer to (see 2.1.1., 2.1.2 and 2.1.3).

Reviewer 2 Report

The authors propose a protocol for biomonitoring occupational exposure and effects of e-waste. The manuscript is well written and the protocol is well described. However, some minor revisions are required before publication. 

Specific remarks:

The protocol should also include buccal swipe and hair samples from the control groups. This will allow for the investigators a better coverage for all the analytes across the different biological matrices in each group.

The protocol should include blood, hair, and buccal swipe after day 1 of work. This will allow for the investigation to observe a cumulative exposure.  This will lead to a better understanding of the biological endpoint since some effects are exposure threshold-dependent.

The protocol should also include the measurement of melanin in the hair since it is well established that certain metals bind to melanin which will affect their bio disponibility. 

The protocol should include DNA extraction from the blood samples for assessing genetic polymorphisms. Based on the observed biomarkers of effects, the investigator may need to draw an association with certain genetic polymorphisms.

Typos:

Page 5 line 197: The word For is repeated.

Page 5 line 219: The title Phthalates would be 2.1.3

Page 5 line 233: The word via should be italicized.

Author Response

The authors propose a protocol for biomonitoring occupational exposure and effects of e-waste. The manuscript is well written and the protocol is well described. However, some minor revisions are required before publication.

Specific remarks:

  1. The protocol should also include buccal swipe and hair samples from the control groups. This will allow for the investigators a better coverage for all the analytes across the different biological matrices in each group.

Author’s response:

It was our intention to collect buccal swipe and hair samples from the control group as well. This applies to all the biological samples that will be collected. Table 3 has been modified to indicate this more clearly.

  1. The protocol should include blood, hair, and buccal swipe after day 1 of work. This will allow for the investigation to observe a cumulative exposure. This will lead to a better understanding of the biological endpoint since some effects are exposure threshold-dependent.

Author’s response: This study protocol describes a cross-sectional study design. Therefore, we do not plan repeated sample collection over time. Only if we expect a background at the beginning of the work week, we will collect a pre-shift sample on the first day like e.g. in the case of chromium in urine. For other biomarkers we do not expect meaningful changes to occur within one workweek. In line 251 we have clarified that measurements will be done during one workweek.

  1. The protocol should also include the measurement of melanin in the hair since it is well established that certain metals bind to melanin which will affect their bio disponibility.

Author’s response:

The reviewer is right that metals bind differently to melanin, but this will not necessary constitute a problem in our study protocol since we will have matching controls. Also, we have a short questionnaire for hair sampling were details about the hair color and used treatments are given. From an analytical point of view, the disponiblity of metals is not an issue, since the samples will be digested using microwave acid digestion. Furthermore, in the recent years the influence of melanin as well as sex and age, have been shown to be minimum either for metals or for organic compounds biomonitoring in hair for environmental exposure or workers (Izydorczyk, Grzegorz et al., 2021; Liang, G., Pan, L., & Liu, X., 2017; Appenzeller et al., 2017). We have therefore not made changes in the revision related to this comment.

  1. The protocol should include DNA extraction from the blood samples for assessing genetic polymorphisms. Based on the observed biomarkers of effects, the investigator may need to draw an association with certain genetic polymorphisms.

Author’s response: The focus of our study is on exposure assessment and primary prevention of exposure. Assessment of genetic polymorphisms would be more relevant for a study aiming at assessment of the health risk.

  1. Page 5 line 197: The word For is repeated.

Author’s response:

This has been corrected in the revision

  1. Page 5 line 219: The title Phthalates would be 2.1.3

Author’s response:

We have corrected this section number in the revision

  1. Page 5 line 233: The word via should be italicized.

Author’s response:

We have adopted this suggestion in the revision.

Round 2

Reviewer 1 Report

The authors addressed all of my points sufficiently and I recommend publishing this manuscript.